# Toward Generalizability of Graph-based Imputation on Biomedical Tabular-based Missing Data

## Abstract

Recent advances in graph-based imputation methods for addressing missing data have received considerable attention, primarily for their ability to effectively aggregate and propagate information through graph structures. However, the applicability of these methods to the biomedical tabular domain remains constrained by two main factors: the lack of task-relevant graph structure and a lack of consideration of feature-wise relationships. To address these challenges, we introduce GRASS[1], a novel approach that effectively bridges the gap between existing graph-based imputation methods and the unique needs of biomedical tabular domains with initially missing data. To derive *feature gradient*, GRASS initiates with training a Multi-Layer Perceptron layer on tabular data. This gradient then facilitates the creation of graph structures from a feature (column) perspective, enabling column-wise feature propagation for imputing missing values, followed by uncertainty-aware categorical clamping. Finally, to effectively utilize existing graph-based imputation methods in an agnostic manner, we input a so-called warmed-up matrix along with an associated sample (row) graph. We validate GRASS on real-world biomedical tabular datasets, demonstrating its ability to unleash the potential of graph-based imputation methods across a variety of missing scenarios.

## 1 Introduction

Graph-based imputation (GBI) has significantly advanced the handling of the missing data imputation (MDI) problem and its impact on downstream tasks like classification in both graph Taguchi et al. (2021); Jiang & Zhang (2020); Rossi et al. (2021) and tabular You et al. (2020); Zhong et al. (2023) domains. Its key advantage lies in the ability to aggregate information from neighboring samples, offering a substantial improvement over traditional methods that predominantly utilize statistical techniques to exploit the distribution of non-missing data Efron (1994); Little & Rubin (2019). Despite these advancements, their generalizability in varied domains, especially those with frequent real-world missing data scenarios like the tabular-based biomedical domain, remains underexplored mainly due to the following two challenges:

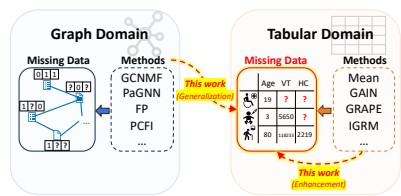

Figure 1: Distant from existing works that are tailored to each specific domain, this work focuses on the generalizability and enhancement of current graph-based and tabular-based imputation methods, with a special focus on the *biomedical* tabular domain.

**Lack of task-relevant graph structure.** The challenge in the tabular domain, in contrast to the well-researched graph domain, stems from a lack of task-relevant graph structures. This lack hinders the application of graph-based methods effectively used in domains where such structures are readily available and well-understood. To corroborate our argument, we compared representatitve methods from graph and tabular domain. As graph structure is absent in the tabular domain, we created a widely-used, sample-wise, similarity-based $k$NN graph from a zero-imputed feature matrix to adapt GBI methods on the graph domain. Figure 2 (a) illustrates, in citation networks, where natural graph structures exist, advanced GBI methods like GCNMF Taguchi et al. (2021) that is based on a Gaussian

---

[1]It stands for general terms, **Gra**ph and Mi**ss**ing. Source code is provided at the Supplemetary Material.

Figure 2: Classification performance comparison between graph and tabular biomedical domain. Blue and Orange represents graph- and tabular-based methods, respectively. In the Graph domain, GCNMF outperforms Mean, leveraging graph structure. In the Bio domain, GCNMF lags behind the Mean baseline until it meets our proposed method, GRASS. In the Medical domain, a recent tabular method, IGRM, underperforms compared to GCNMF but achieves similar results with GRASS. For GCNMF, a cosine-similarity-based $k$NN graph is utilized for the graph structure. Notably, graph datasets typically involve a manually set missing ratio (MR), as they are fully observed initially. In contrast, the biomedical domain naturally encounters an initial missing ratio (IMR), reflecting more practical settings.

Mixture Model significantly outperform traditional tabular imputation techniques. However, in the tabular such as biomedical domain, the situation is quite different. Factors like high dimensionality coupled with small sample sizes, technical limitations such as dropout Wiens (2003), and patient data confidentiality issues Cismondi et al. (2013b) intensify the difficulties of handling missing data. These unique factors make deriving a relevant graph structure particularly challenging in scenarios with high rates of missing data. As depicted in Figure 2 (b), utilizing a $k$NN graph generated from an initial feature matrix proves inadequate for methods like GCNMF underperforming even basic approaches like Mean. The limitation arises because the graph depends on an incomplete feature matrix that lacks crucial task-relevant information, impacting both effective imputation and subsequent downstream tasks. This motivates us to develop a more sophisticated graph structure enriched with task-relevant information.

**Lack of consideration of column-wise relationships.** Specifically, in the biomedical tabular domain, characterized by complex interactions between various features like gene-gene and disease-related interactions and high dimensionality, neglecting feature relationships can be a critical oversight. For instance, the relationship between 'Age' and 'Ventricles' (VT), indicating potential brain volume loss with age, is crucial in biomedical analysis, as referenced in studies Nestor et al. (2008); Bjork et al. (2003). Overlooking such feature (i.e., column-wise) relationships in data can cause advanced graph-structure generating methods like IGRM, which uses a bipartite graph with a row-wise approach, to be less effective than simpler, row-wise methods like GCNMF. This is evident in Figure 2 (c), highlighting the importance of integrating relevant feature relationships in the development of graph structure, especially in the tabular domain.

Given these considerations, the central question arises:

> *(Q) Is it feasible to craft a more insightful feature matrix and associated graph structure, thereby leveraging the potential of graph-based imputation in the biomedical domain?*

Here, we introduce GRASS, an innovative approach that offers an orthogonal way to leverage and generalize existing graph-based imputation methods to real-world missing scenarios—such as those in the biomedical tabular domain—where both initially missing features and graph structure are prevalent. Instead of directly constructing a graph structure based on the current incomplete features, which would be suboptimal, we commence with training on the tabular data using the Multi-Layer Perceptron (MLP) layer. During this training process, a valuable task-relevant byproduct that naturally emerges is *gradient information with respect to the features*. We utilize this feature gradient by concatenating it to the original feature matrix, thereby creating a feature perspective graph. Following this, we implement column-wise feature propagation to impute the initial feature matrix and apply uncertainty-aware categorical clamping, preserving the uncertain status for later imputation by graph-based imputation techniques. Now, equipped with this so-called warmed-up feature matrix and the new graph structure, we stand ready to harness the potential of cutting-edge graph-based imputation techniques, extending our reach to real-world missing data scenarios. Figure 1 visually summarizes the core contributions of our work.

In summary, our contributions are three-fold:

- We, for the first time, explore the generalizability of recent graph-based imputation models in the context of real-world biomedical tabular data with missing values.

- We propose a novel approach for constructing a graph structure that incorporates feature gradient information.

- We demonstrate that GRASS can serve as an effective initial starting point in a model-agnostic fashion, thereby enhancing performance in downstream tasks across multiple biomedical datasets.

## 2 RELATED WORK

**Tabular-based Data Imputation.** The challenge of missing data imputation has a long history and many early approaches for tabular data are rooted in statistical methods Efron (1994); Little & Rubin (2019). These methods often leverage the distribution of non-missing values to impute missing ones. Recent machine learning-based imputation techniques include kNN-based approaches Troyanskaya et al. (2001); Keerin et al. (2012), GAIN Yoon et al. (2018), which employs Generative Adversarial Networks Goodfellow et al. (2020), and MIWAE Mattei & Frellsen (2019), which utilizes a Deep Latent Variable model Kingma & Welling (2013). There have also been efforts to adapt graph structures to tabular data for imputation; for example, GRAPE You et al. (2020) introduces a bipartite graph connecting samples and features, while the more recent IGRM Zhong et al. (2023) extends GRAPE by adding a friend network to capture relationships between samples. However, as these methods heavily rely on the input feature matrix as a main resource, in cases where a significant proportion of data is missing, the imputation quality tends to degrade, negatively affecting downstream tasks' performance. Notably, compared to the graph domain, most of these studies emphasize either imputation or regression. This is because the task of imputing continuous values closely aligns with regression, simplifying both training and evaluation. However, another pivotal downstream task, *i.e.*, classification, remains underexplored in the realm of tabular data with missing features.

**Graph-based Data Imputation.** From the viewpoint of graph-based imputation, GCNMF Taguchi et al. (2021) tackles missing features by assuming a Gaussian distribution for each feature channel while aligning it with Graph Convolutional Networks (GCN) Kipf & Welling (2016a). PaGNN Jiang & Zhang (2020) proposes a partial aggregation scheme derived from neighborhood reconstruction. FP Rossi et al. (2021) iteratively diffuses known features to unknown features, followed by GNN layers. Recently, PCFI Um et al. (2023) builds upon FP to introduce channel-wise diffusion confidence to handle scenarios with higher missing feature rates. However, channel-wise diffusion operates on fully connected graphs, potentially incorporating irrelevant or noisy information between channels. Additionally, they carry a strong inductive bias toward readily available graph structures, limiting their generalizability. As mentioned above, the application domain of these works primarily focuses on Citation Sen et al. (2008) and Co-Purchase networks Shchur et al. (2018) where features are text-based, a situation less reflective of realistic cases where features are initially missing.

**Biomedical Data Imputation.** In the medical domain, several research efforts have been made to address missing data. Multiple imputation techniques are suggested by Janssen et al. (2010), while Cismondi et al. (2013a) employs statistical approaches for imputation. The MICE algorithm Van Buuren & Groothuis-Oudshoorn (2011) is also widely applied in this context. On the biology side, a prominent issue related to missing data is the occurrence of dropout events in single-cell RNA-sequencing datasets, where zero values are often falsely recorded as missing. Among the various methods proposed van Dijk et al. (2018); Wang et al. (2021); Yun et al. (2023), scGNN Wang et al. (2021) and scFP Yun et al. (2023) employ Graph Auto-Encoders (GAE) Kipf & Welling (2016b) and FP Rossi et al. (2021) to impute these false zeros. Despite these efforts, the use of graph-based data imputation techniques remains underexplored. This is largely due to the absence of a network structure and a reliance on input feature matrices. Such limitations widen the gap between recent advances in graph-based imputation and real-world applications where data is often missing.

## 3 METHOD

In this section, we introduce GRASS, a novel algorithm designed to bridge the gap between recent graph-based imputation methods and biomedical missing data. Initially, we employ a Multi-Layer

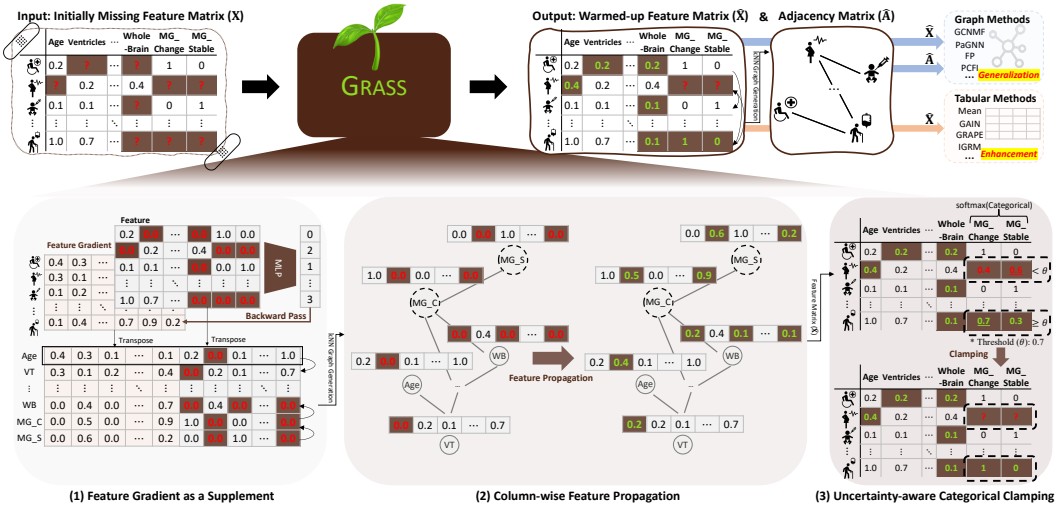

Figure 3: Overall framework of GRASS. Given an initially missing feature matrix, we first train a simple MLP to obtain the feature gradient. By concatenating these gradients with the initial matrix, we create a graph from a feature-wise perspective. After employing Column-wise Feature Propagation, followed by Uncertainty-aware Categorical Clamping, we obtain a warmed-up feature matrix and an adjacency matrix. These serve as the foundational feature and adjacency matrices for existing graph-based imputation methods.

Perceptron (MLP) to extract the feature gradient, a crucial supplement for graph structure used for imputation (Sec 3.1). Subsequently, we implement a Column-wise Feature Propagation grounded on the gradient-informed graph (Sec 3.2). Next, we apply uncertainty-aware categorical clamping (Sec 3.3), leading to the creation of a warmed-up matrix and an adjacency matrix. These matrices then become the inputs for existing graph-based imputation methods. The comprehensive framework of GRASS is illustrated in Figure 3, while the detailed algorithm of the entire process GRASS can be found in Appendix A.2.

**Task: Classification with Tabular Data Containing Initial Missing Features.** Given an initially missing feature matrix $\mathbf{X} \in \mathbb{R}^{N \times F}$, where $N$ denotes the total samples and $F$ the feature dimensions, the goal of GRASS is to produce a warmed-up feature matrix accompanied by a sample-wise graph structure. These enhanced matrices enable existing graph-based imputation methods to seamlessly utilize them as an initial reference point.

## 3.1 FEATURE GRADIENT AS A SUPPLEMENT

Given *initially missing* feature matrix, the direct utilization of this matrix for downstream tasks can lead to suboptimal results where prior imputation is imperative. Naturally, one might consider the latest graph-based imputation techniques Taguchi et al. (2021); Jiang & Zhang (2020); Rossi et al. (2021); Um et al. (2023) given their prowess in receiving messages from neighboring samples. However, as shown in Figure 2 (b), constructing a graph structure directly from a partially observed feature matrix and then proceeding with imputation is extremely challenging, often yielding inferior results than simple tabular-based methods. In this context, given our primary goal is the downstream task, i.e., classification, we choose not to rely on the initial partially observed feature matrix. Instead, we leverage the additional resource, supervision signal, incorporating this information into graph construction. To achieve this, we employ a simple Multi-Layer Perceptron (MLP) to capture and utilize the *feature gradient*, acquired during backpropagation, as a crucial, task-aligned resource. These gradients indicate how subtle shifts in features impact the model's predictions, highlighting the salience of individual features in loss minimization. By supplementing this gradient, we can devise a graph structure that encapsulates not just observed feature information from an initial state but also the feature saliency in relation to our targeted downstream task. We start with a formal definition of a feature gradient.

**Definition 3.1.** *The feature gradient, denoted as $\boldsymbol{\nabla}_{\mathbf{X}}$, represents the partial derivatives of the loss function concerning each feature in the input matrix and is mathematically defined as $\boldsymbol{\nabla}_{\mathbf{X}} = \frac{\partial \mathcal{L}}{\partial \mathbf{X}}$.*

Building upon **Definition 3.1**, we derive feature gradient through the training of a straightforward MLP [2]. During this training process, the ensuing proposition emerges:

**Proposition 3.2.** *Consider a 2-layer Multi-Layer Perceptron (MLP). The output for each layer is formulated as: $\mathbf{Z}' = \sigma(\mathbf{X}\mathbf{W}' + \mathbf{b}'), \mathbf{Z}'' = \mathbf{Z}'\mathbf{W}'' + \mathbf{b}''$ where the trainable weight matrices are denoted as $\mathbf{W}' \in \mathbb{R}^{F \times D}$ and $\mathbf{W}'' \in \mathbb{R}^{D \times C}$, and bias vectors are represented by $\mathbf{b}' \in \mathbb{R}^D$ and $\mathbf{b}_2 \in \mathbb{R}^C$. The activation function, $\sigma$, is chosen as the ReLU function, $F$ is the feature dimension, and $D$ specifies the dimension. Upon applying the softmax function, we derive the prediction probability matrix $\hat{\mathbf{Y}} \in \mathbb{R}^{N \times C}$, with $C$ indicating the number of classes. $\mathbf{Y} \in \mathbb{R}^{\mathbf{N} \times \mathbf{C}}$ is a label matrix. Using cross-entropy as the loss function, the feature gradient, represented as $\boldsymbol{\nabla}_{\mathbf{X}} \in \mathbb{R}^{N \times F}$, can be computed as:*

$$\boldsymbol{\nabla}_{\mathbf{X}} = ((\hat{\mathbf{Y}} - \mathbf{Y}) \cdot \mathbf{W}''^{\top}) \odot (\mathbf{X}\mathbf{W}' + \mathbf{b}' > 0) \cdot \mathbf{W}'^{\top}$$

Please refer to Appendix A.1 for the detailed proof.

A central observation from **Proposition 3.2** is the dynamic nature of the feature gradient matrix across MLP training epochs, despite the static nature of the initially provided missing feature matrix $\mathbf{X}$. This dynamic is attributed to task-favorable adjustments in trainable weight parameters (e.g., $\mathbf{W}', \mathbf{W}''$), which in turn influence feature gradient variations. Here, considering feature gradients undergo a change at every epoch, persistently storing these gradients across epochs incurs substantial memory overhead, $\mathcal{O}(NF)$. Moreover, there's no guarantee of consistent gradient quality improvement with each epoch. To address this, we selectively store feature gradient[3] only when the MLP's performance on predicting the validation set improves, leveraging them as pivotal cues to enhance downstream task efficacy. In essence, after training MLP, we consolidate the stacked feature gradient, averaging them to yield $\overline{\boldsymbol{\nabla}}_{\mathbf{X}} \in \mathbb{R}^{N \times F}$, a matrix accordant in shape with the original feature matrix. It is important to note that while the calculation of the feature gradient may appear complex, in practice, the gradient can be easily obtained by activating the gradient-saving switch, as shown in Appendix A.3.

## 3.2 COLUMN-WISE FEATURE PROPAGATION

In classification scenarios, the resulting imputed matrices usually experience sample-wise (row-wise) message-passing, particularly when using GNNs as classifiers. However, this row-wise adjacency matrix approach might not capture crucial relationships because it inherently assumes feature channel independence. This aspect holds particular importance in biomedical domains, e.g., Alzheimer's disease, exemplified by the relationship between 'Age' and 'Ventricles' in the Introduction. Hence, before establishing a row-wise graph, we prioritize the creation of a column-wise graph structure, which provides an opportunity to encapsulate intra-feature relationships. Considering that the initial columns (i.e., features) have missing values, we address this challenge by a supplement, the feature gradient we derived earlier, as follows:

$$\mathbf{A}^{feat} = k_{\text{col}}\text{-nearest-neighbor}(\overline{\boldsymbol{\nabla}}_{\mathbf{X}}^{\top} \| \mathbf{X}^{\top}) \tag{1}$$

where $k_{\text{col}}$-nearest-neighbor($\cdot$) denotes the connection of $k_{\text{col}}$ neighbors for each feature channel, established using cosine similarity, with $k_{\text{col}}$ as a hyperparameter. Given the feature-wise graph, we employ FP Rossi et al. (2021) to estimate missing features across iterations by capturing inter-feature relationships in a *column-wise fashion* while preserving known values, which is depicted as follows:

$$\mathbf{X}^{(i+1)\top} = \tilde{\mathbf{A}}^{feat}\mathbf{X}^{(i)\top},$$
$$\mathbf{X}^{(i+1)\top}_{v,d} = \mathbf{X}^{(0)\top}_{v,d}, \forall v \in \mathcal{V}_{known,d}, \forall d \leq F \tag{2}$$

---

[2]During MLP training, we employed zero imputation for initially missing values to leverage its computational efficiency and flexibility. This approach avoids the assumption that missing data occurs completely at random (MCAR), a condition often not met in the biomedical domain.

[3]L2-normalization was applied during feature gradient storage to maintain consistent feature scales and retain the original vector's directionality.

where $\tilde{\mathbf{A}}^{feat} = \mathbf{D}^{-1/2}\mathbf{A}^{feat}\mathbf{D}^{-1/2} \in \mathbb{R}^{F \times F}$ is symmetrically normalized adjacency, having cosine similarity as a weight, with a self-loop with added degree matrix $\mathbf{D}$. At iteration $i$, the matrix is represented as $\mathbf{X}^{(i)\top} \in \mathbb{R}^{F \times N}$. The set $\mathcal{V}_{known,d}$ contains nodes with known feature values for the $d$-th channel. After $K$ iterations and another transposition, we obtain the imputed output $\hat{\mathbf{X}} = \mathbf{X}^{(K)} \in \mathbb{R}^{N \times F}$, which we term the *warmed-up matrix*. Considering that our approach utilizes a custom $k$NN graph, as opposed to the pre-defined adjacency matrix used in FP, detailed discussions on convergence can be found in Appendix A.4.

### 3.3 UNCERTAINTY-AWARE CATEGORICAL CLAMPING

In real-world tabular datasets, which often include both numerical (e.g., Age, Blood Pressure) and categorical features (e.g., Gender, Blood Type), we introduce a clamping method specifically designed for categorical features. Previous column-wise FP, involving iterative multiplications with a normalized adjacency matrix, can yield continuous imputed values for one-hot encoded categorical columns. In the biomedical field, particularly in Alzheimer's Disease (AD) research, the treatment of categorical features, such as the status of Microglia (MG) cells, is of great importance Hansen et al. (2018). The presence or absence of MG cell changes (denoted as 1 for 'MG_Change' and 0 for 'MG_Stable') can be a significant indicator of the disease's progression, requiring meticulous consideration. Practitioners might classify an MG status as 'change' only if the imputed value surpasses a predefined threshold, set at 0.7 in this context. Naturally, imputed values below this threshold will be categorized as 'MG_Stable'.

However, recall that our imputation has, until now, solely considered column-wise relationships. Therefore, we choose to leave room for uncertain values by retaining their original missing status ('?'). This approach opens up the possibility for subsequent row-wise propagation using existing graph-based imputation methods, thereby allowing for potentially higher-value imputation later on. Formally, for a categorical index $c$, which we obtain during the preprocessing of numerical and categorical mixed type tabular data, with corresponding bin count $c_b$ for the original column, the predicted probability vector for a sample $j$ from the continuous imputed matrix is given by: $\tilde{\mathbf{x}}_c = \mathrm{softmax}(\hat{\mathbf{X}}_{j,c:c+c_b}) \in \mathbb{R}^{c_b}$ Subsequently, the clamping process is as below:

$$\hat{\mathbf{X}}_{j,c:c+c_b} = \begin{cases} \mathrm{OneHot}(\mathrm{argmax}(\tilde{\mathbf{x}}_c)), & \text{if } \max(\tilde{\mathbf{x}}_c) \geq \theta \\ \underbrace{[?, \ldots, ?]}_{c_b \text{ times}}, & \text{otherwise} \end{cases} \tag{3}$$

where $\mathrm{OneHot}(\cdot)$ function represents one-hot encoding based on a threshold, $\theta$. The symbol $?$ indicates retained initial missing values, emphasizing our aim to preserve inherent uncertainties.

**GRASS as an Initializer.** Given warmed-up feature matrix $\hat{\mathbf{X}}$, we proceed to construct a row-wise (sample-wise) graph structure defined as $\hat{\mathbf{A}} = k_{\text{row}}\text{-nearest-neighbor}(\hat{\mathbf{X}})$. Here, $k_{\text{row}}$-nearest-neighbor$(\cdot)$ establishes connections among $k_{\text{row}}$ neighbors for each sample, utilizing cosine similarity, with $k_{\text{row}}$ serving as a hyperparameter. Equipped with these matrices, namely $\hat{\mathbf{X}}$ and $\hat{\mathbf{A}}$, we are now ready to enjoy the potentials of any existing GBI methods through a model-agnostic fashion.

## 4 EXPERIMENTS

**Datasets.** We evaluate GRASS on nine datasets, each initially containing missing data. Four of these datasets are from the bio single-cell RNA-seq domain: Mouse ES Klein et al. (2015), Pancreas Luecken et al. (2022), Baron Human Baron et al. (2016), and Mouse Bladder Han et al. (2018). The remaining five datasets are from the medical domain: Breast Cancer Zwitter & Soklic (1988), Hepatitis hep (1988), Duke Breast Saha et al. (2018), ADNI Petersen et al. (2010), and ABIDE Di Martino et al. (2014). Dataset splits were randomly generated with five different train/val/test divisions, with a ratio of 10%, 10%, 80%. Comprehensive details and statistics for each dataset are available in Appendix A.5. In terms of evaluation metrics, we use Macro-F1 scores for the bio domain while employing AUROC scores for the medical domain.

Table 1: Pancreas.

| | OG | + GRASS init. | Impr. (%) |
|---|---|---|---|
| | Pancreas (IMR: 56.65%) | | |
| LP | $0.656_{\pm 0.039}$ | $\mathbf{0.798}_{\pm 0.068}$ | 21.66 |
| GCNMF | $0.527_{\pm 0.210}$ | $\underline{0.708}_{\pm 0.087}$ | 34.27 |
| PaGNN | $0.701_{\pm 0.044}$ | $0.768_{\pm 0.040}$ | 9.58 |
| Zero | $0.687_{\pm 0.066}$ | $0.783_{\pm 0.062}$ | 14.02 |
| NM | $0.679_{\pm 0.047}$ | $0.788_{\pm 0.068}$ | 16.09 |
| FP | $0.716_{\pm 0.046}$ | $0.788_{\pm 0.068}$ | 10.08 |
| PCFI | $0.673_{\pm 0.055}$ | $0.686_{\pm 0.040}$ | 1.95 |
| Mean | $0.616_{\pm 0.044}$ | $0.619_{\pm 0.032}$ | 0.44 |
| kNN | $0.652_{\pm 0.047}$ | $0.706_{\pm 0.048}$ | 8.34 |
| GAIN | $0.638_{\pm 0.075}$ | $0.738_{\pm 0.024}$ | 15.66 |
| MIWAE | OOM | - | - |
| GRAPE | OOM | - | - |
| IGRM | OOM | - | - |
| scFP | $0.743_{\pm 0.044}$ | $0.788_{\pm 0.085}$ | 6.05 |

Table 2: Baron Human.

| | OG | + GRASS init. | Impr. (%) |
|---|---|---|---|
| | Baron Human (IMR: 57.25%) | | |
| LP | $0.736_{\pm 0.022}$ | $0.828_{\pm 0.055}$ | 12.46 |
| GCNMF | $0.350_{\pm 0.130}$ | $\underline{0.817}_{\pm 0.066}$ | 133.30 |
| PaGNN | $0.777_{\pm 0.043}$ | $0.820_{\pm 0.057}$ | 5.53 |
| Zero | $0.812_{\pm 0.030}$ | $0.842_{\pm 0.049}$ | 3.71 |
| NM | $0.758_{\pm 0.045}$ | $0.801_{\pm 0.084}$ | 5.71 |
| FP | $0.789_{\pm 0.039}$ | $0.802_{\pm 0.084}$ | 1.61 |
| PCFI | $0.769_{\pm 0.036}$ | $0.792_{\pm 0.038}$ | 2.96 |
| Mean | $0.672_{\pm 0.010}$ | $0.694_{\pm 0.023}$ | 3.41 |
| kNN | $0.746_{\pm 0.048}$ | $0.760_{\pm 0.053}$ | 1.82 |
| GAIN | $0.728_{\pm 0.041}$ | $0.745_{\pm 0.033}$ | 2.39 |
| MIWAE | OOM | - | - |
| GRAPE | OOM | - | - |
| IGRM | OOM | - | - |
| scFP | $0.809_{\pm 0.067}$ | $\mathbf{0.853}_{\pm 0.0.031}$ | 5.43 |

Table 3: Mouse Bladder.

| | OG | + GRASS init. | Impr. (%) |
|---|---|---|---|
| | Mouse Bladder (IMR: 69.05%) | | |
| LP | $0.556_{\pm 0.030}$ | $0.643_{\pm 0.053}$ | 15.57 |
| GCNMF | $0.300_{\pm 0.182}$ | $\underline{0.701}_{\pm 0.042}$ | 133.90 |
| PaGNN | $0.713_{\pm 0.056}$ | $\mathbf{0.775}_{\pm 0.028}$ | 8.78 |
| Zero | $0.712_{\pm 0.015}$ | $0.768_{\pm 0.031}$ | 7.83 |
| NM | $0.721_{\pm 0.050}$ | $0.775_{\pm 0.030}$ | 7.38 |
| FP | $0.686_{\pm 0.046}$ | $0.772_{\pm 0.036}$ | 12.48 |
| PCFI | $0.710_{\pm 0.046}$ | $0.727_{\pm 0.028}$ | 2.41 |
| Mean | $0.555_{\pm 0.074}$ | $0.569_{\pm 0.062}$ | 2.39 |
| kNN | $0.587_{\pm 0.038}$ | $0.674_{\pm 0.059}$ | 14.82 |
| GAIN | $0.585_{\pm 0.045}$ | $0.649_{\pm 0.038}$ | 10.84 |
| MIWAE | OOM | - | - |
| GRAPE | OOM | - | - |
| IGRM | OOM | - | - |
| scFP | $0.653_{\pm 0.024}$ | $0.759_{\pm 0.022}$ | 16.23 |

Table 4: Breast Cancer.

| | OG | + GRASS init. | Impr. (%) |
|---|---|---|---|
| | Breast Cancer (IMR: 0.35%) | | |
| LP | $0.561_{\pm 0.038}$ | $0.562_{\pm 0.041}$ | 0.14 |
| GCNMF | $0.551_{\pm 0.033}$ | $\mathbf{0.579}_{\pm 0.049}$ | 5.02 |
| PaGNN | $0.540_{\pm 0.037}$ | $0.562_{\pm 0.032}$ | 3.98 |
| Zero | $0.542_{\pm 0.048}$ | $0.557_{\pm 0.039}$ | 2.71 |
| NM | $0.538_{\pm 0.049}$ | $\underline{0.566}_{\pm 0.052}$ | 5.07 |
| FP | $0.543_{\pm 0.047}$ | $0.565_{\pm 0.052}$ | 4.05 |
| PCFI | $0.545_{\pm 0.039}$ | $0.547_{\pm 0.040}$ | 0.44 |
| Mean | $0.562_{\pm 0.045}$ | $0.562_{\pm 0.045}$ | 0.00 |
| kNN | $0.552_{\pm 0.041}$ | $0.556_{\pm 0.041}$ | 0.67 |
| GAIN | $0.566_{\pm 0.044}$ | $0.567_{\pm 0.043}$ | 0.21 |
| MIWAE | $0.558_{\pm 0.033}$ | $0.563_{\pm 0.035}$ | 0.93 |
| GRAPE | $0.572_{\pm 0.029}$ | $0.573_{\pm 0.017}$ | 0.26 |
| IGRM | $0.548_{\pm 0.039}$ | $0.552_{\pm 0.037}$ | 0.66 |
| scFP | $0.554_{\pm 0.047}$ | $0.563_{\pm 0.055}$ | 1.62 |

Table 5: Hepatitis.

| | OG | + GRASS init. | Impr. (%) |
|---|---|---|---|
| | Hepatitis (IMR: 5.67%) | | |
| LP | $0.573_{\pm 0.078}$ | $0.608_{\pm 0.053}$ | 6.06 |
| GCNMF | $0.685_{\pm 0.097}$ | $0.707_{\pm 0.088}$ | 3.22 |
| PaGNN | $0.729_{\pm 0.074}$ | $\mathbf{0.741}_{\pm 0.058}$ | 1.74 |
| Zero | $0.713_{\pm 0.090}$ | $0.714_{\pm 0.088}$ | 0.14 |
| NM | $0.702_{\pm 0.071}$ | $0.702_{\pm 0.071}$ | 0.00 |
| FP | $0.705_{\pm 0.085}$ | $0.707_{\pm 0.092}$ | 0.20 |
| PCFI | $0.728_{\pm 0.108}$ | $0.728_{\pm 0.108}$ | 0.00 |
| Mean | $0.691_{\pm 0.072}$ | $0.711_{\pm 0.081}$ | 2.86 |
| kNN | $0.612_{\pm 0.097}$ | $0.626_{\pm 0.105}$ | 2.15 |
| GAIN | $0.578_{\pm 0.093}$ | $\underline{0.646}_{\pm 0.080}$ | 11.63 |
| MIWAE | $0.573_{\pm 0.080}$ | $0.608_{\pm 0.077}$ | 6.25 |
| GRAPE | $0.701_{\pm 0.033}$ | $0.706_{\pm 0.032}$ | 0.63 |
| IGRM | $0.668_{\pm 0.087}$ | $0.703_{\pm 0.109}$ | 5.26 |
| scFP | $0.691_{\pm 0.077}$ | $0.691_{\pm 0.077}$ | 0.00 |

Table 6: ABIDE.

| | OG | + GRASS init. | Impr. (%) |
|---|---|---|---|
| | ABIDE (IMR: 69.74%) | | |
| LP | $0.894_{\pm 0.009}$ | $0.895_{\pm 0.011}$ | 0.13 |
| GCNMF | $0.819_{\pm 0.042}$ | $0.913_{\pm 0.010}$ | 11.49 |
| PaGNN | $0.907_{\pm 0.009}$ | $0.914_{\pm 0.008}$ | 0.82 |
| Zero | $0.902_{\pm 0.008}$ | $0.915_{\pm 0.008}$ | 1.38 |
| NM | $0.905_{\pm 0.011}$ | $\mathbf{0.918}_{\pm 0.007}$ | 1.48 |
| FP | $0.908_{\pm 0.014}$ | $0.915_{\pm 0.005}$ | 0.86 |
| PCFI | $0.915_{\pm 0.008}$ | $0.917_{\pm 0.010}$ | 0.26 |
| Mean | $0.607_{\pm 0.027}$ | $\underline{0.905}_{\pm 0.007}$ | 49.09 |
| kNN | $0.896_{\pm 0.009}$ | $0.907_{\pm 0.010}$ | 1.16 |
| GAIN | $0.793_{\pm 0.010}$ | $0.910_{\pm 0.009}$ | 14.70 |
| MIWAE | $0.623_{\pm 0.015}$ | $0.898_{\pm 0.008}$ | 44.10 |
| GRAPE | $0.889_{\pm 0.010}$ | $0.906_{\pm 0.006}$ | 1.90 |
| IGRM | $0.747_{\pm 0.019}$ | $0.908_{\pm 0.004}$ | 21.54 |
| scFP | $0.894_{\pm 0.010}$ | $0.903_{\pm 0.007}$ | 1.00 |

**Compared Methods.** To verify whether our algorithm enhances current graph-based imputation methods, we compare it with established baselines such as Label Propagation (LP) Zhu (2005), GCNMF Taguchi et al. (2021), PaGNN Jiang & Zhang (2020), Neighborhood Mean (NM) Rossi et al. (2021), Zero Imputation with GCN layers (Zero) Rossi et al. (2021), FP Rossi et al. (2021), and PCFI Um et al. (2023). Given our focus on tabular data, we also include common methods like Mean Little & Rubin (2019), kNN Troyanskaya et al. (2001), GAIN Yoon et al. (2018), MI-WAE Mattei & Frellsen (2019), and recent graph-based approaches like GRAPE You et al. (2020) and IGRM Zhong et al. (2023). Detailed explanations for each method are provided in Appendix A.6. To benchmark against a domain-specific baseline, we included scFP Yun et al. (2023). Since scFP also employs the FP method, a thorough comparison between scFP and GRASS is provided in Appendix A.7. The detailed hyperparameter setting is provided in Appendix A.8.

## 4.1 CLASSIFICATION PERFORMANCE

Tables 1, 2, and 3 show classification performance in the bio domain, while Tables 4, 5, and 6 do so for the medical domain. Key observations include: **1)** GBI methods like FP and PCFI excel tabular-based methods, largely due to their message-passing mechanisms. **2)** Despite this, as shown in Tables 2 and 4, GBI methods such as GCNMF and PaGNN occasionally fall short of basic tabular data methods, a trend also observed in Figure 1. However, their integration with GRASS's warmed-up matrix and adjacency matrix notably enhances their performance. It is also important to note that the performance gain in the bio domain is more pronounced. This is attributed to a higher initial missing ratio and the domain's relative simplicity, as it consists only of numerical features. In contrast, the medical domain, which includes a mix of numerical and categorical features, presents more complex challenges. **3)** While FP and PCFI generally outperform GCNMF in citation networks with high missing rates, as demonstrated in each paper, GCNMF shows significant potential in the biomedical domain, especially when paired with the right graph structure and Gaussian Mixture Model, as evidenced by a 133.90% improvement in the Mouse Bladder dataset (Table 3). **4)** Although primarily designed for a more well-defined adjacency matrix, tabular-based methods like Mean and GAIN also benefit from using the warmed-up matrix, suggesting its utility as an effective starting point across models. More performance on other datasets can be found in Appendix A.9. In summary, while the

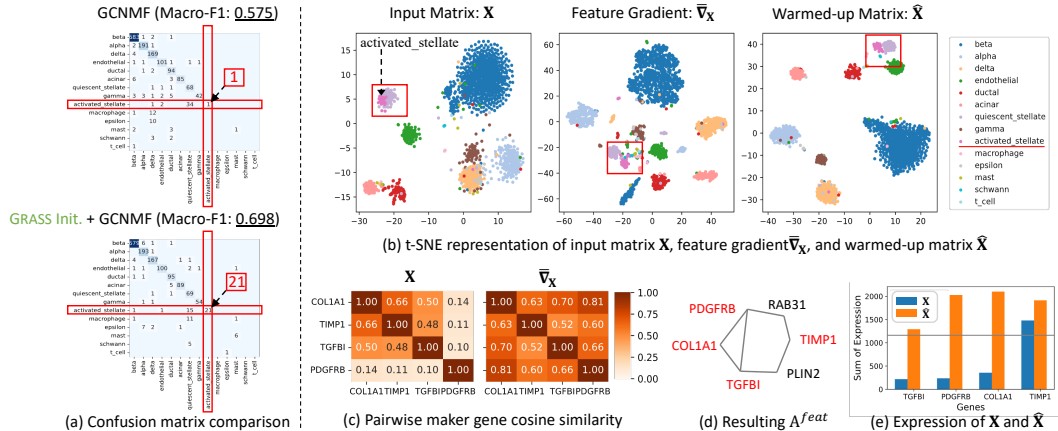

Figure 4: Exploring the influence of feature gradient in Pancreas dataset. (a) Confusion matrix comparison between original GCNMF and its GRASS initialized version, illustrating the latter capturing more rare cell-type. (b) t-SNE representation of $\mathbf{X}$, $\overline{\nabla}_{\mathbf{X}}$, $\hat{\mathbf{X}}$. 'activated stellate' cell type is represented as pink color. (c) Pairwise marker gene cosine similarity comparison between original feature matrix ($\mathbf{X}$ and feature gradient($\nabla_{\mathbf{X}}$), resource for the column-wise graph. (d) Resulting $\mathbf{A}^{feat}$ via utilizing feature gradient as a supplement. (e) Expression of four marker genes being amplified after column-wise Feature Propagation. All experiments were conducted on the Pancreas dataset. Marker genes, which are key factors for classifying 'activated stellate' cell type, were identified based on existing research linking these genes to the activated hepatic stellate cell (HSC).

generalizability of current GBI models has not been thoroughly explored in the biomedical domain, incorporating GRASS initially can maximize their capabilities.

## 4.2 WHY FEATURE GRADIENT MATTERS?

In Figure 4, we delve into the contribution of our key component, 'feature gradient' on addressing the issue of missing features. Figure 4 (a) presents a comparative analysis between the GCNMF model and its GRASS-initialized counterpart, particularly highlighting their performance in the Pancreas dataset. A notable aspect of this comparison is the enhanced accuracy in classifying rare cell types such as 'activated stellate'. From Figure 4 (b), we observe that this improvement is largely due to the feature gradient, which provides more distinct class representations compared to the original input matrix, as indicated by the t-SNE representation of the warmed-up matrix obtained through column-wise FP. The final warmed-up matrix, enriched with feature gradients, shows an enhanced intra-class distribution, especially for the 'activated stellate' cell type. This suggests that feature gradient plays a crucial supplement role in learning more distinct class representations. To confirm our observations, we analyzed marker genes for 'activated stellate' as depicted in Figure 4 (c). This investigation revealed that marker genes such as COL1A1, TIMP1, TGFBI, and PDGFRB demonstrate higher cosine similarity in the feature gradient compared to their expressions in the original matrix. This was achievable owing to its ability to incorporate task-relevant information, i.e., 'activated stellate' cell-type information, which is brought from the label supervision. By leveraging this task-relevant gradient information, in Figure 4 (d), direct connections (i.e., 1-hop neighbors) have been formed among three marker genes, while one displays 2-hop relationships. Following column-wise feature propagation, an increase in the expression levels of marker genes due to neighborhood aggregation is observed, as shown in Figure 4 (e). The observed increase in gene expression is significant as it exceeds the average expression of all genes, marked by the gray line. This indicates that the increase occurs in a cell-type-specific manner, highlighting the targeted and precise nature of the gene expression changes. Consequently, the enhanced expression of the marker gene for the 'activated stellate' cell type plays a significant role in identifying rare cell types, a task that was previously unachievable without the integration of feature gradients.

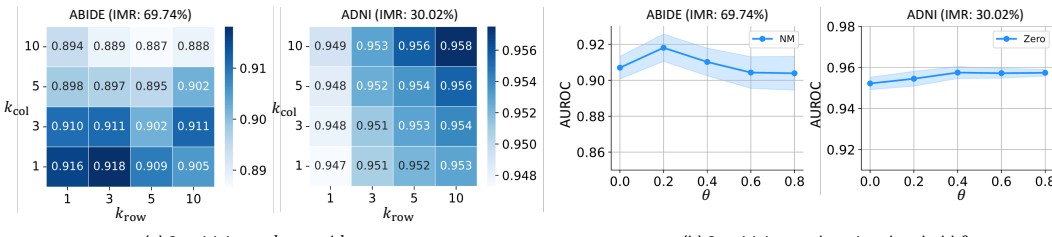

(a) Sensitivity on $k_{\text{col}}$ and $k_{\text{row}}$    (b) Sensitivity on clamping threshold $\theta$

Figure 5: Sensitivity analysis on hyperparameters for GRASS. AUROC is measured in both datasets.

### 4.3  ABLATION & SENSITIVITY STUDIES

Table 7 presents an ablation study of GRASS components, revealing three key insights: **1)** Combining both row- and column-wise feature propagation, as GRASS does by first executing column-wise FP then row-wise propagation, shows clear benefits. **2)** In column-wise FP, incorporating feature gradients enhances performance, underscoring their importance in constructing column-wise graphs. **3)** In medical domains with mixed data types, the clamping technique effectively maintains the original scale of data, with its strategy of retaining original missing values due to uncertainties proving most effective. Additionally, Table 8 supports the improved structure of the GRASS warmed-up adjacency matrix, $\hat{\mathbf{A}}$, over the initial matrix, $\mathbf{A}$. This finding aligns with our initial goal of crafting a more insightful graph structure for biomedical data, demonstrating the advantages of GRASS in graph-based imputation methods. Figure 5 (a) shows the sensitivity of hyperparameters $k_{\text{col}}$ and $k_{\text{row}}$, responsible for edge generation in column- and row-wise graphs. GRASS demonstrates robustness within the recommended range $\{1, 3, 5, 10\}$. However, in datasets with high missing rates like ABIDE (69.74%), using a

Table 7:  Ablation study of GRASS. Here, two best-performing models, GCNMF and PaGNN, are used for the backbone model. UaCC stands for Uncertainty-aware Categorical Clamping. (w/o room) implies that uncertain values are not left as missing but are instead imputed. The last row corresponds to GRASS.

| Model Variants | Breast Cancer | Hepatitis |
|---|---|---|
| Row only | $0.500\pm0.00$ | $0.667\pm0.10$ |
| Col only | $0.524\pm0.08$ | $0.603\pm0.20$ |
| Col+$\nabla_{\mathbf{X}}$ | $0.540\pm0.10$ | $0.627\pm0.11$ |
| Col+$\nabla_{\mathbf{X}}$+UaCC (w/o room) | $0.577\pm0.05$ | $0.736\pm0.07$ |
| Col+$\nabla_{\mathbf{X}}$+UaCC (w room) | $0.579\pm0.08$ | $0.742\pm0.06$ |

Table 8: Edge homophily ratio comparison between original adjacency matrix with refined adjacency matrix obtained via GRASS. Edge homophily ratio: $\frac{\text{number of edges connecting two nodes with same labels}}{\text{number of total edges}}$

| | $\mathbf{A}$ | $\hat{\mathbf{A}}$ | Impr. (%) |
|---|---|---|---|
| Mouse ES | 0.8591 | 0.9900 | 15.24 |
| Pancreas | 0.9319 | 0.9819 | 5.37 |
| Baron Human | 0.9557 | 0.9788 | 2.42 |
| Mouse Bladder | 0.5672 | 0.8046 | 41.86 |
| Breast Cancer | 0.6698 | 0.6701 | 0.05 |
| Hepatitis | 0.7902 | 0.8035 | 1.68 |
| Duke Breast | 0.6887 | 0.7074 | 2.72 |
| ADNI | 0.7130 | 0.7336 | 2.89 |
| ABIDE | 0.9142 | 0.9166 | 0.26 |

larger $k$ in column-wise graphs could lead to over-smoothing issues. The impact of higher $k$ values is further discussed in Appendix A.4. Figure 5 (b) underlines the importance of an appropriate clamping threshold. A larger $\theta$ preserves more uncertainties, while a smaller $\theta$ may lead to early imputation, impacting further refinement by subsequent imputation methods. Details on complexity and further extensions are in Appendices A.10 and A.11.

## 5  CONCLUSION

Graph-based imputation methods, increasingly popular for filling missing features by leveraging neighborhood information, face challenges in the biomedical tabular domain due to the absence of task-relevant graph structures and intra-feature relationship considerations. We introduce GRASS, an innovative algorithm designed to generalize and enhance graph-based imputation to the biomedical domain. GRASS starts with obtaining feature gradients to construct a column-wise graph, followed by feature propagation and uncertainty-aware categorical clamping. Our extensive research validates the effectiveness of GRASS, positioning it as a promising foundation for future graph-based imputation research in biomedical domains.

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

# A APPENDIX

## A.1 PROOF OF PROPOSITION 1

**Proposition 1.** *Consider a 2-layer Multi-Layer Perceptron (MLP). The output for each layer is formulated as: $\mathbf{Z}' = \sigma(\mathbf{XW}' + \mathbf{b}'), \mathbf{Z}'' = \mathbf{Z}'\mathbf{W}'' + \mathbf{b}''$ where the trainable weight matrices are denoted as $\mathbf{W}' \in \mathbb{R}^{F \times D}$ and $\mathbf{W}'' \in \mathbb{R}^{D \times C}$, and bias vectors are represented by $\mathbf{b}' \in \mathbb{R}^D$ and $\mathbf{b}_2 \in \mathbb{R}^C$. The activation function, $\sigma$, is chosen as the ReLU function, $F$ is the feature dimension, and $D$ specifies the dimension. Upon applying the softmax function, we derive the prediction probability matrix $\hat{\mathbf{Y}} \in \mathbb{R}^{N \times C}$, with $C$ indicating the number of classes. $\mathbf{Y} \in \mathbb{R}^{\mathbf{N} \times \mathbf{C}}$ is a label matrix. Using cross-entropy as the loss function, the feature gradient, represented as $\boldsymbol{\nabla}_{\mathbf{X}} \in \mathbb{R}^{N \times F}$, can be computed as:*

$$\boldsymbol{\nabla}_{\mathbf{X}} = ((\hat{\mathbf{Y}} - \mathbf{Y}) \cdot \mathbf{W}''^{\top}) \odot (\mathbf{XW}' + \mathbf{b}' > 0) \cdot \mathbf{W}'^{\top}$$

*Proof.* Given a row-vector, $\mathbf{x} \in \mathbb{R}^{1 \times F}$, consider the following application of the chain rule:

$$\frac{\partial \mathcal{L}}{\partial \mathbf{x}} = \frac{\partial \mathcal{L}}{\partial \mathbf{z}''} \cdot \frac{\partial \mathbf{z}''}{\partial \mathbf{z}'} \cdot \frac{\partial \mathbf{z}'}{\partial \mathbf{x}}$$

To compute $\frac{\partial \mathcal{L}}{\partial \mathbf{z}''}$, let's begin by considering a specific class index $n$, when $n$ ranges from 1 to $C$, the total number of classes.

$$\frac{\partial \mathcal{L}}{\partial z_n''} = \frac{\partial \mathcal{L}}{\partial \hat{y}_n} \cdot \frac{\hat{y}_n}{\partial z_n''}$$

$$= -\sum_{i=1}^{C} y_i * \frac{\partial \log(\hat{y}_i)}{\partial \hat{y}_i} * \frac{\partial \hat{y}_i}{\partial z_n''} = -\sum_{i=1}^{C} \frac{y_i}{\hat{y}_i} * \frac{\partial \hat{y}_i}{\partial z_n''}$$

To determine $\frac{\partial \hat{y}_i}{\partial z_n''}$, the gradient with respect to the softmax function for each class $i$ in total $C$ classes can be computed:

I. When $i = n$,

$$\frac{\partial \hat{y}_i}{\partial z_i''} = \frac{\partial}{\partial z_i''} \left( \frac{e^{z_i''}}{\sum_{j=1}^{C} e^{z_j''}} \right)$$

$$= \frac{e^{z_i''} * \sum_{j=1}^{C} e^{z_j''} - \left( e^{z_i''} \right)^2}{\left( \sum_{j=1}^{C} e^{z_j''} \right)^2}$$

$$= \frac{e^{z_i''}}{\sum_{j=1}^{C} e^{z_j''}} * \frac{\sum_{j=1}^{C} e^{z_j''} - e^{z_i''}}{\sum_{j=1}^{C} e^{z_j''}}$$

$$= \hat{y}_i * (1 - \hat{y}_i)$$

II. When $i \neq n$,

$$\frac{\partial \hat{y}_i}{\partial z_n''} = \frac{0 * \sum_{j=1}^{C} e^{z_j''} - e^{z_i''} * e^{z_n''}}{\left(\sum_{j=1}^{C} e^{z_j''}\right)^2}$$

$$= -\frac{e^{z_i''} * e^{z_n''}}{\left(\sum_{j=1}^{C} e^{z_j''}\right)^2}$$

$$= -\frac{e^{z_i''}}{\sum_{j=1}^{C} e^{z_j''}} * \frac{e^{z_n''}}{\sum_{j=1}^{C} e^{z_j''}}$$

$$= -\hat{y}_i * \hat{y}_n$$

We can subsequently consolidate two separate cases as follows:

$$\frac{\partial L}{\partial z_n''} = -\sum_{i=1}^{C} \frac{y_i}{\hat{y}_i} * \frac{\partial \hat{y}_i}{\partial z_n''}$$

$$= -\frac{y_n}{\hat{y}_n} * \hat{y}_n * (1 - \hat{y}_n) + \sum_{i \neq n}^{c} \frac{y_i}{\hat{y}_i} * \hat{y}_i * \hat{y}_n$$

$$= -y_n + y_n * \hat{y}_n + \sum_{i \neq n}^{c} y_i * \hat{y}_n$$

$$= -y_n + \sum_{i=1}^{C} y_i * \hat{y}_n$$

$$= \hat{y}_n - y_n$$

The vector form for the same is:

$$\frac{\partial \hat{\mathbf{y}}}{\partial \mathbf{z}''} = \hat{\mathbf{y}} - \mathbf{y}$$

Now, the gradient with respect to the output of the hidden layer, $\frac{\partial \mathbf{z}''}{\partial \mathbf{z}'}$ is directly given by:

$$\frac{\partial \mathbf{z}''}{\partial \mathbf{z}'} = \mathbf{W}''^\top$$

Lastly, to obtain $\frac{\partial \mathbf{z}'}{\partial \mathbf{x}}$, we need to consider the ReLU activation in the hidden layer:

$$\frac{\partial \mathbf{z}'}{\partial \mathbf{x}} = \frac{\partial \mathbf{z}'}{\partial \sigma(\mathbf{x}\mathbf{W}' + \mathbf{b}')} \cdot \frac{\partial \sigma(\mathbf{x}\mathbf{W}' + \mathbf{b}')}{\partial \mathbf{x}} = (\mathbf{x}\mathbf{W}' + \mathbf{b}' > 0) \cdot \mathbf{W}'^\top$$

Combining these results yields the feature gradient in row-vector ($\mathbb{R}^{1 \times F}$) format:

$$\frac{\partial \mathcal{L}}{\partial \mathbf{x}} = \frac{\partial \mathcal{L}}{\partial \mathbf{z}''} \cdot \frac{\partial \mathbf{z}''}{\partial \mathbf{z}'} \cdot \frac{\partial \mathbf{z}'}{\partial \mathbf{x}}$$

$$= ((\hat{\mathbf{y}} - \mathbf{y}) \cdot \mathbf{W}''^\top) \odot (\mathbf{x}\mathbf{W}' + \mathbf{b}' > 0) \cdot \mathbf{W}'^\top$$

where $\odot$ represents the element-wise multiplication (Hadamard product).

When generalized for the entire dataset, the matrix ($\mathbb{R}^{N \times F}$) format becomes:

$$\boldsymbol{\nabla}_{\mathbf{X}} = \frac{\partial \mathcal{L}}{\partial \mathbf{X}} = \frac{\partial \mathcal{L}}{\partial \mathbf{Z}''} \cdot \frac{\partial \mathbf{Z}''}{\partial \mathbf{Z}'} \cdot \frac{\partial \mathbf{Z}'}{\partial \mathbf{X}}$$

$$= ((\hat{\mathbf{Y}} - \mathbf{Y}) \cdot \mathbf{W}''^\top) \odot (\mathbf{X}\mathbf{W}' + \mathbf{b}' > 0) \cdot \mathbf{W}'^\top$$

$\square$

## A.2 PSEUDOCODE OF GRASS

Algorithm 1 presents the pseudocode for our proposed algorithm, GRASS. By training the MLP, we derive the feature gradient, which is utilized to generate a column-wise graph (see line 3). We then execute Column-wise Feature Propagation (line 5) and clamp the categorical columns (line 6). Consequently, we produce the warmed-up feature matrix and the adjacency matrix, which will seamlessly align with existing graph-based imputation methods.

## A.3 OBTAINING FEATURE GRADIENT IN PRACTICE

Here, we provide a PyTorch-style pseudocode in Listing 1, detailing the function for obtaining the feature gradient (corresponds to line 15 in Algorithm 1). In training the 2-layer MLP, as shown in Line 22, we activate the 'requires_grad' attribute by setting it to True. This enables AutoGrad in PyTorch to automatically calculate the feature gradient following backpropagation, a value that is then accessible in Line 28. It is crucial to note that there is no update to the original feature matrix; it remains static, with only the classifier's weights being updated. This process dynamically alters the value of the feature gradient through these modified weights, as demonstrated in Proposition 3.2. Additionally, as indicated in Line 37, we save the feature gradient only when there is an improvement in validation performance, which is an efficient approach to memory usage. After training the MLP, which typically involves early stopping, we compute the average of the gradients to obtain the final feature gradient.

---

**Algorithm 1** Pseudocode of the proposed algorithm

---

1: **Input:** Initial missing feature matrix $\mathbf{X}$, train label matrix $\mathbf{Y}$
2: **Output:** Warmed-up feature matrix $\hat{\mathbf{X}}$, adjacency matrix $\hat{\mathbf{A}}$
3: $\overline{\boldsymbol{\nabla}}_{\mathbf{X}} \leftarrow$ TrainMLP$(\mathbf{X}, ValidationSet)$
4: $\mathbf{A}_{feat} \leftarrow k_{\text{col}}$-nearest-neighbor$(\overline{\boldsymbol{\nabla}}_{\mathbf{X}}^{\top} \| \mathbf{X}^{\top})$
5: $\mathbf{X}^{(K)^{\top}} \leftarrow$ Propagation$(\mathbf{A}_{feat}, \mathbf{X}^{(0)\top}, \mathcal{V}_{known}, K)$
6: $\hat{\mathbf{X}} \leftarrow$ Clamper$\mathbf{X}^{(K)\top}$
7: $\hat{\mathbf{A}} \leftarrow k_{\text{row}}$-nearest-neighbor$(\hat{\mathbf{X}})$
8: **function** TrainMLP$(\mathbf{X}, ValidationSet)$
9:     Initialize highest validation performance as $V_{\text{highest}} = 0$
10:    Initialize empty list $G = []$
11:    **while** not converged **do**
12:       Train MLP for one epoch using training data
13:       Compute validation performance $V_{\text{current}}$
14:       **if** $V_{\text{current}} > V_{\text{highest}}$ **then**
15:          $\boldsymbol{\nabla}_{\mathbf{X}} \leftarrow ((\hat{\mathbf{Y}} - \mathbf{Y}) \cdot \mathbf{W}''^{\top}) \odot (\mathbf{X}\mathbf{W}' + \mathbf{b}' > 0) \cdot \mathbf{W}'^{\top}$
16:          Append the $\boldsymbol{\nabla}_{\mathbf{X}}$ to list $G$
17:          Update $V_{\text{highest}} \leftarrow V_{\text{current}}$
18:       **end if**
19:    **end while**
20:    $\overline{\boldsymbol{\nabla}}_{\mathbf{X}} \leftarrow \frac{1}{\text{length}(G)} \sum_{g \in G} g$
21:    **return** $\overline{\boldsymbol{\nabla}}_{\mathbf{X}}$
22: **end function**
23: **function** Propagation$(\mathbf{A}, \mathbf{W}, Known, K)$
24:    $\mathbf{M} \leftarrow \mathbf{W}$
25:    **for** $k \leftarrow 1$ to $K$ **do**
26:       $\mathbf{W} \leftarrow \mathbf{A}\mathbf{W}$
27:       $\mathbf{W}_{Known} \leftarrow \mathbf{M}_{Known}$
28:    **end for**
29:    **return** $\mathbf{W}$
30: **end function**
31: **function** Clamper$(\hat{\mathbf{X}})$
32:    **for** $i \leftarrow 1$ to $N$ **do**
33:       **for** $j \leftarrow c$ to length$(CategoricalColumns)$ **do**
34:          $\tilde{\mathbf{x}}_c \leftarrow \text{softmax}(\hat{\mathbf{X}}_{j,c:c+c_b})$
35:    
$$\hat{\mathbf{X}}_{j,c:c+c_b} = \begin{cases} \text{OneHot}(\text{argmax}(\tilde{\mathbf{x}}_c)), & \text{if } \max(\tilde{\mathbf{x}}_c) \geq \theta \\ [\underbrace{?, \ldots, ?}_{c_b \text{ times}}], & \text{otherwise} \end{cases}$$
36:       **end for**
37:    **end for**
38:    **return** $\hat{\mathbf{X}}$
39: **end function**

---

```python
def obtain_feature_gradient(
            x, # missing feature matrix
            classifier, # 2-layer MLP
            labels, # supervisions
            train_mask,
            val_mask,
            epochs
            )

    # Initialize missing features as zeros
    x = torch.nan_to_num(x, 0)

    optimizer = optim.Adam(classifier.parameters())
    best_val_performance = 0
    grads = []

    for epoch in range(0, epochs):
        classifier.train()
        optimizer.zero_grad()

        # Allow tracking gradients for x
        x.requires_grad=True
        out = classifier(x)
        loss = F.CrossEntropy(out[train_mask], labels[train_mask])

        loss.backward()
        optimizer.step()

        grad = x.grad # Feature Gradient
        x.requires_grad=False

        classifier.eval()
        out = classifier(x)

        val_performance = roc_auc_score(out[val_mask], labels[val_mask])

        # Save gradient
        if best_val_performance <= val_performance:
            best_val_performance = val_performance
            grads.append(F.normalize(grad, dim=0, p=2).cpu())

    # Average gradients
    feature_gradient = torch.mean(torch.stack(grads), dim=0)

    return feature_gradient
```

Listing 1: PyTorch-style pseudocode for obtaining feature gradient via training 2-layer MLP.

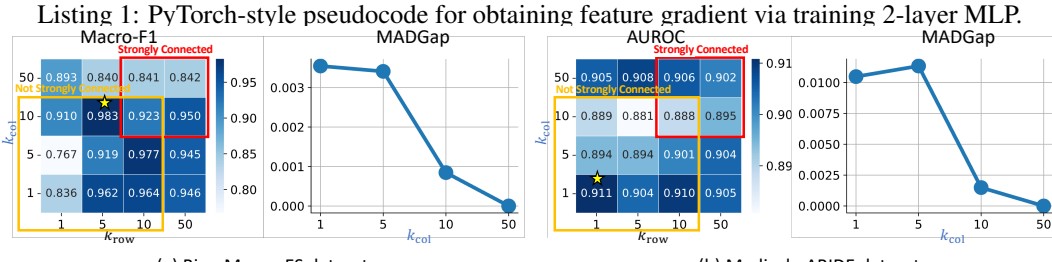

(a) Bio - Mouse ES dataset        (b) Medical - ABIDE dataset

Figure 6: Performance comparison upon increasing the values of $k_{col}$ and $k_{row}$, which are responsible for generating the column-wise and row-wise graphs, respectively. This increase ensures convergence in (a) the Mouse ES dataset and (b) a Medical dataset. For each dataset, the best-performing models, FP and NM, with GRASS initialized, are utilized to assess performance. The MADGap metric, calculated as the normalized distance in the warmed-up matrix ($\hat{\mathbf{X}}$) between remote nodes within an 8-hop distance and neighboring nodes within a 3-hop distance (as suggested in the original paper), is used to measure oversmoothing. A smaller MADGap value indicates a more severe oversmoothing.

## A.4 DISCUSSION ON CONVERGENCE OF COLUMN-WISE FEATURE PROPAGATION

One of the hallmark advantages of FP is its ability to guarantee convergence of feature representations for missing nodes, provided the graph is undirected and maintains strong connectivity (Berman & Plemmons, 1994). In contrast to graph domains where the initial graph structure is given without any missing elements and can thereby extract a strongly connected component, our situation, defined by initially missing features devoid of a graph structure, requires manual graph construction, such as the $k$NN graph as detailed in Equation 2. This approach does not ensure strong connectivity, making the convergence of imputed values for missing features uncertain. Nonetheless, we argue that within our context of missing features, simply increasing the number of neighbors, $k$, to achieve the convergence property might not always be advantageous.

*Claim:* Elevating $k$ to attain strong connectivity (which increases the likelihood, albeit without guarantees) and consequently secure the convergence property can sometimes be detrimental to performance. This might inadvertently introduce a primary drawback inherent to graph-based learning: over-smoothing. $\Leftrightarrow$ *Rationale:* As the value of $k$ escalates, the adjacency matrix $\mathbf{A}^{\text{feat}}$ becomes increasingly dense. However, considering our scenario of missing features where feature representation remains incomplete, the veracity of the new connections becomes dubious. For the representation of missing nodes in the feature matrix used in Equation 2, denoted as $\mathbf{X}^{\top} \in \mathbb{R}^{F \times N}$ and represented by $\mathbf{x}_u \in \mathbb{R}^N$, a high missing rate combined with an extensive $k$ implies that the feature representation of the majority node, $\mathbf{x}_u$, will evolve via feature propagation. As the number of layers increases and $k$ approaches the total number of nodes $F$, these nodes end up with almost identical representations.

Given this perspective, we aim to avert ambiguous node connections and counteract over-smoothing, which could potentially degrade classification performance. To this end, we commit to using a relatively modest and smaller value of $k$ when crafting the graph from the feature's perspective.

**Discussion on the Convergence and Performance Gain Relationship.** To further investigate whether the convergence property contributes to performance gain, we conducted an empirical analysis to validate our claims. In Figure 6, we extended our proposed range of $k_{\text{col}}$ and $k_{\text{row}}$ values, $\{1,3,5,10\}$, up to 50, and tested the resulting graph's connectivity. We observed that when $k_{\text{col}}$ and $k_{\text{row}}$ exceed 10, the generated graph becomes strongly connected, meaning that every node is reachable from every other node. Interestingly, while strong connectivity provides convenience in choosing the number of neighbors and satisfies the necessary condition for FP to converge, it *does not necessarily translate to performance gains*. Optimal performance was, in fact, achieved within a smaller range of $k$ values, as initially proposed. Upon further investigation, we discovered that increasing $k_{\text{col}}$ leads to an *oversmoothing issue* in the resulting output, particularly in the warmed-up matrix. This effect was quantified using the MADGap metric (Chen et al., 2020), which measures the representational difference between remote and neighboring nodes. In summary, our findings suggest that when dealing with bio-medical tabular data, where an initial graph structure is not provided and a $k$NN graph must be manually generated, selecting a large $k$ value to leverage the convergence property of FP may not be the most effective strategy in scenarios with severe missing.

## A.5 DETAILS OF DATASETS

In the bio datasets, we make use of cell-gene matrices to predict the relevant annotated cell types for each cell. This cell type information serves as the supervisory signal during training. For preprocessing, we typically filter out cells and genes that have not been transcribed in each row and column, respectively, and apply a log transformation to normalize the count values.

- **Mouse ES** (Klein et al., 2015) dataset employs a droplet-microfluidic approach for parallel barcoding. We used concatenated data originally separated by different days post-leukemia inhibitory factor (LIF) withdrawal, treating the day of withdrawal as the annotation for the cell type.
- **Pancreas** (Luecken et al., 2022) dataset, obtained via the inDrop method, captures the transcriptomes of individual pancreatic cells from four human donors and two mouse strains. It includes 14 annotated cell types.
- **Baron Human** (Baron et al., 2016) dataset focuses on individual pancreatic cells from human donors, sequenced using a droplet-based method. It features 14 annotated cell types.

- **Mouse Bladder** (Han et al., 2018) dataset, sourced from the Mouse Cell Atlas (MCA) project and sequenced via the Microwell-seq platform, includes cell types as defined by the original authors' annotations.

In our medical datasets, we focused on datasets that originally include missing values and feature a mix of categorical and numerical features. During preprocessing, we removed rows and columns if all features were missing in each sample or if all samples were missing in each feature, respectively. We selected the most representative feature column related to the patient's diagnosis as the class label for prediction.

- **Breast Cancer** (Asuncion & Newman, 2007): Published in the UCI repository and provided by the Oncology Institute, this dataset contains tumour-related features. We use 'recurrence', a binary attribute, as the class label.

- **Hepatitis** (Asuncion & Newman, 2007): Also published in the UCI repository, this dataset includes data on hepatitis occurrences in individuals, with attributes related to liver characteristics. The binary annotation of the patient's outcome (die or live) is used as the class label.

- **Duke Breast** (Saha et al., 2018): Made available by The Cancer Imaging Archive (TCIA), this dataset consists of medical images and non-image clinical data for tumor prediction. From the tabular data provided, we use the 'Tumor_Grade' feature, which indicates the grade of the tumor, as the class label.

- **ADNI** (Petersen et al., 2010): This collection includes various types of medical images and non-image clinical data related to Alzheimer's disease. We utilize the 'DX_bl' feature from the clinical data, indicating the patient's diagnosis, as the class label.

- **ABIDE** (Di Martino et al., 2014): Containing data on autism spectrum disorder based on brain imaging and clinical data, this dataset uses the 'DX_Group' feature from the clinical data, which represents the diagnostic group of the patient, as the class label.

Table 9 provides an overview of dataset statistics. In the medical domain, where features can be both numerical and categorical, we employed MinMaxScaler for numerical columns and one-hot encoding for categorical ones. For the bio domain, we employed datasets from the single-cell RNA-sequencing domain. In this domain, both false-zeros and biologically true zeros coexist (van Dijk et al., 2018; Li & Li, 2018). However, since we cannot distinguish whether a given zero is a false-zero or a true-zero, we treat this situation as a missing data scenario. Accordingly, we consider zeros as missing values, aligning with the approach taken in the recent work, scFP (Yun et al., 2023). The initial missing ratio (IMR) represents the absence of data in the original table before any preprocessing. The final column of Table 9 indicates the extent of missing data even after obtaining the warmed-up feature matrix and adjacency matrix. This phenomenon is particularly evident in datasets with categorical features. Yet, the designed allowance for subsequent graph-based imputation methods has proven to complement effectively, as illustrated in Table 7. The dataset split of train/validation/test as 10:10:80 is particularly relevant given the shift in the scRNA-seq domain. Traditionally, this domain has been approached through unsupervised methods. However, the growing availability of public scRNA-seq datasets and known cell types has increasingly steered research towards supervised machine learning models. This evolution in research methodology reflects the changing landscape and emerging trends in the field, as noted in recent studies Cao et al. (2022).

Table 9: Statistics of datasets. (IMR: Initially Missing Rate)

| Dataset | Domain | $N$ | $F$ | Num. | Cat. | Preprocessed | $C$ | IMR | GRASS Init. |
|---|---|---|---|---|---|---|---|---|---|
| Mouse ES | Bio | 2717 | 24047 | 24047 | 0 | 2000 | 4 | 27.21% | 0.00% |
| Pancreas | Bio | 1937 | 15575 | 15575 | 0 | 2000 | 14 | 56.65% | 0.00% |
| Baron Human | Bio | 8569 | 17499 | 17499 | 0 | 2000 | 14 | 57.25% | 0.00% |
| Mouse Bladder | Bio | 2746 | 19771 | 19771 | 0 | 2000 | 16 | 69.05% | 0.14% |
| Breast Cancer | Medical | 286 | 9 | 1 | 8 | 39 | 2 | 0.35% | 0.00% |
| Hepatitis | Medical | 155 | 19 | 4 | 15 | 298 | 2 | 5.67% | 5.53% |
| Duke Breast | Medical | 907 | 93 | 34 | 59 | 3364 | 3 | 11.94% | 9.42% |
| ADNI | Medical | 2419 | 113 | 92 | 21 | 2741 | 5 | 30.02% | 4.18% |
| ABIDE | Medical | 1112 | 72 | 64 | 8 | 284 | 2 | 69.74% | 3.39% |

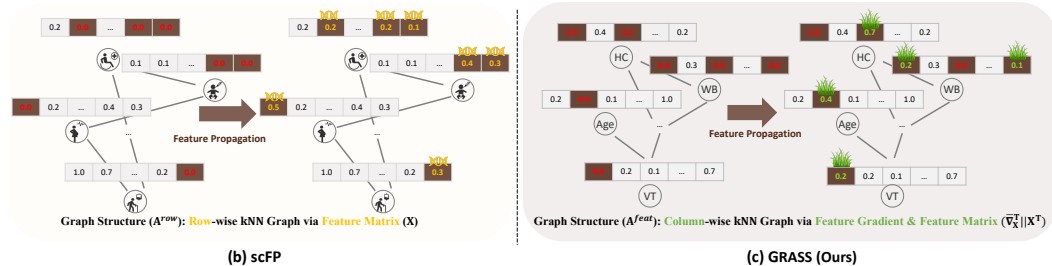

Figure 7: Comparison of scFP and GRASS. (a) scFP builds a Row-wise $k$NN graph only via input feature matrix ($\mathbf{X}$). (b) GRASS builds a Column-wise $k$NN graph incorporating both the input feature matrix ($\mathbf{X}$) and the supplementary feature gradient ($\overline{\nabla}_{\mathbf{X}}$).

## A.6 DETAILS OF BASELINES

To tackle the challenge of generalizing graph-based imputation methods to bio-medical tabular data, we have adopted two types of baseline approaches. For graph-based imputation methods, which typically target downstream tasks like classification, we adopted widely-used methods as follows.

- **LP** (Zhu, 2005) is a semi-supervised algorithm that spreads known labels to similar data points in an unlabeled dataset, based on the given graph structure.
- **GCNMF** (Taguchi et al., 2021) is an end-to-end GNN-based model that imputes missing features by assuming a Gaussian Mixture Model aligned with GCN.
- **PaGNN** (Jiang & Zhang, 2020) is a GNN-based method that implements a partial message-passing scheme, propagating only observed features.
- **Zero** (Rossi et al., 2021) is a simple 2-layer Graph Convolution Network. We impute missing features with zeros in this model.
- **NM** (Rossi et al., 2021) imputes missing features by averaging the features of one-hop neighboring nodes, followed by GCN layers.
- **FP** (Rossi et al., 2021) propagates given features through neighbors, replacing observed ones with their original values to minimize Dirichlet energy.
- **PCFI** (Um et al., 2023) improves FP by considering the relationship among features with pseudo confidence, defined by the shortest path to the known feature.

In our experiments, we initially used zero imputation for missing values when applying these methods. Additionally, we explored hybrid approaches that combine elements of tabular and graph-based methods, including MEAN and the more recent tabular baseline, IGRM. However, we found that these methods did not perform optimally, as they both struggled in scenarios with initial missing data. For the classifier, we utilized a 2-layer Graph Convolutional Network (GCN) as our classifier. Additionally, as our primary focus is on tabular data, we include common table-based imputation methods as follows.

- **Mean** (Little & Rubin, 2019) replaces missing values in a dataset with the mean value of the available data for the same feature.
- **kNN** (Troyanskaya et al., 2001) imputes missing data by finding the $k$ nearest neighbors based on cosine similarity and then averaging their features.
- **GAIN** (Yoon et al., 2018) uses a generative adversarial network to impute missing values, where one network generates candidates and another evaluates them.
- **MIWAE** (Mattei & Frellsen, 2019) employs a type of autoencoder for multiple imputations, capturing the data's underlying distribution to provide multiple plausible values for missing data.
- **GRAPE** (You et al., 2020) adopts a bipartite graph framework, viewing observations and features as two node types, and imputes missing values through edge-level prediction.

- **IGRM** (Zhong et al., 2023) enhances the bipartite graph framework by introducing the concept of a friend network, which denotes relationships between samples.

In these methods, which were originally designed for imputing missing values, a logistic classifier has been incorporated to perform classification tasks.

## A.7 COMPARISON BETWEEN SCFP AND GRASS

As GRASS integrates FP with the aim of enhancing generalizability in the bio-medical domain, it is necessary to compare it with the recently proposed single-cell Feature Propagation (scFP), which also adopts FP, specifically targeting the single-cell RNA-seq domain.

- **(1) Target Domain**: While scFP focuses on the scRNA-seq domain, particularly from a biological perspective, GRASS adopts a more general approach for the broader 'biomedical' domain, as indicated in the paper's title. This distinction is crucial as scRNA-seq datasets typically comprise *numerical features* where each element represents the count of a gene's RNA transcript sequenced by the sequencing machine. In contrast, medical datasets often include *both numerical and categorical features*, such as patient information. This versatility underscores the broader applicability of GRASS, capable of handling both numerical and categorical features, the latter through the clamping technique as discussed in Section 3.3. Therefore, we argue that the target domain of scFP, primarily focused on numerical matrix imputation in scRNA-seq, differs from that of GRASS, which extends to handling categorical data often encountered in patient data.

- **(2) Target Task and Imputation Methodology**: Unlike scFP, which is *unsupervised* with its primary goal being effective imputation in sparse and noisy cell-gene count matrices, this work concentrates on *supervised* tasks, specifically on downstream applications like *classification*. Notably, the objective of imputation is often to enhance performance in relevant downstream tasks (Rossi et al., 2021; van Dijk et al., 2018; Wang et al., 2021). In this context, while the unsupervised approach of scFP can align with supervised tasks through probing (i.e., attaching a classifier), it is important to note that since its imputation occurs prior to probing, scFP cannot incorporate any downstream task-related knowledge during the imputation process, potentially leading to shortcomings in classification tasks. Conversely, as GRASS is directly designed with downstream tasks in mind, it incorporates knowledge pertinent to these tasks during imputation. This is achieved by utilizing the *feature gradient*, which is obtained during training 2-layer MLP. This fundamental difference in the target task (classification vs. imputation) and the imputation process (incorporating relevant downstream knowledge or not) distinctly sets the two methodologies apart.

- **(3) Usage of FP**: Although both scFP and GRASS employ FP, their applications of this process differ significantly. Specifically, scFP utilizes FP from a row-wise perspective, i.e., focusing on cell-cell relationships while assuming gene-gene relationship independence. Although beneficial for smoothing similar and relevant samples, this approach does not capture interactions between columns (features), which are pivotal in the bio-medical domain. For instance, in scRNA-seq, gene-gene relationships, such as co-expression networks, play a critical role in identifying key regulatory genes or pathways, offering insights into underlying biological or disease mechanisms (Cochain et al., 2018; Chowdhury et al., 2019; Galfre et al., 2021). Acknowledging this, GRASS initially employs column-wise FP to capture potential feature interactions, e.g., gene-gene relationships. It's also noteworthy that GRASS incorporates not only the feature matrix but also the feature gradient relevant to downstream tasks when generating the column-wise $k$NN graph. Consequently, before initiating row-wise (sample-wise) smoothing in the relevant GNN model, GRASS is able to consider feature relationships that scFP does not capture. This distinction is illustrated in Figure 7 and significantly differentiates the two methodologies.

## A.8 DETAILS OF HYPERPARAMETERS

For graph-based imputation methods, we generated a $k$NN graph, selecting $k$ from {1, 3, 5, 10}. Following Rossi et al. (2021), we set a consistent dropout rate of 0.5 and a dimension of 64 across all

methods. For baselines, their own hyperparameters were tuned based on each paper's recommendations. For our model, we explored values for $k_{col}$ and $k_{row}$ within $\{1, 3, 5, 10\}$. The clamping process's threshold, $\theta$, was tested among $\{0.0, 0.2, 0.4, 0.6, 0.8\}$. We set the number of iterations, $K$, to 40, as advised in the FP paper. Tables 10 and 11 detail the optimal hyperparameter settings when GRASS and existing graph-based imputation models are best aligned.

Table 10: Hyperparameter setting of Best Performing models.

| Dataset | Best Performing | $\theta$ | $k_{col}$ | $k_{row}$ | OG | GRASS | Improvement |
|---|---|---|---|---|---|---|---|
| Mouse ES | FP | - | 10 | 5 | 0.900 | 0.983 | 9.17% |
| Pancreas | LP | - | 3 | 3 | 0.656 | 0.799 | 21.66% |
| Baron Human | scFP | - | 1 | 10 | 0.809 | 0.853 | 5.43% |
| Mouse Bladder | PaGNN | - | 3 | 5 | 0.713 | 0.760 | 8.78% |
| Breast Cancer | GCNMF | 0.2 | 3 | 5 | 0.552 | 0.580 | 5.02% |
| Hepatitis | PaGNN | 0.6 | 5 | 1 | 0.729 | 0.742 | 1.74% |
| Duke Breast | GAIN | 0.4 | 5 | 10 | 0.699 | 0.700 | 0.09% |
| ADNI | Zero | 0.4 | 10 | 10 | 0.956 | 0.960 | 0.16% |
| ABIDE | NM | 0.2 | 1 | 1 | 0.905 | 0.919 | 1.48% |

Table 11: Hyperparameter setting of Most Improved models.

| Dataset | Most Improved | $\theta$ | $k_{col}$ | $k_{row}$ | OG | GRASS | Improvement |
|---|---|---|---|---|---|---|---|
| Mouse ES | GCNMF | - | 5 | 1 | 0.525 | 0.973 | 85.31% |
| Pancreas | GCNMF | - | 10 | 1 | 0.527 | 0.708 | 34.27% |
| Baron Human | GCNMF | - | 5 | 3 | 0.350 | 0.818 | 133.30% |
| Mouse Bladder | GCNMF | - | 1 | 3 | 0.300 | 0.702 | 133.90% |
| Breast Cancer | NM | 0.6 | 10 | 10 | 0.539 | 0.565 | 5.07% |
| Hepatitis | GAIN | 0.0 | 1 | 10 | 0.579 | 0.646 | 11.63% |
| Duke Breast | FP | 0.4 | 3 | 5 | 0.661 | 0.689 | 5.07% |
| ADNI | GCNMF | 0.0 | 3 | 10 | 0.898 | 0.945 | 5.25% |
| ABIDE | Mean | 0.2 | 5 | 10 | 0.608 | 0.906 | 49.09% |

## A.9 ADDITIONAL CLASSIFICATION PERFORMANCE

Performance on an additional bio domain dataset, Mouse ES, is detailed in Table 12, while further results for two medical datasets, Duke Breast and ADNI, are available in Tables 13 and 14, respectively. These results further demonstrate the enhancement of graph-based imputation methods when initialized with GRASS. This underscores the significance of employing a task-relevant, warmed-up feature matrix and adjacency matrix for improved performance in these biomedical domains. The comprehensive performance gains achieved by using GRASSas an initializer across all datasets are presented in Figure 8. This figure highlights the performance improvements of the original methods when initialized with GRASS, visually represented by additional grass-colored vertical bars.

Table 12: Bio-Mouse ES.

| | Mouse ES (IMR: 27.21%) | | |
|---|---|---|---|
| | OG | + GRASS init. | Impr. (%) |
| LP | $0.878_{\pm 0.005}$ | $0.979_{\pm 0.003}$ | 11.43 |
| GCNMF | $0.525_{\pm 0.238}$ | $\underline{0.972}_{\pm 0.008}$ | 85.31 |
| PaGNN | $0.899_{\pm 0.072}$ | $0.980_{\pm 0.002}$ | 9.03 |
| Zero | $0.960_{\pm 0.005}$ | $0.982_{\pm 0.004}$ | 2.30 |
| NM | $0.885_{\pm 0.098}$ | $0.982_{\pm 0.004}$ | 10.99 |
| FP | $0.900_{\pm 0.100}$ | $\mathbf{0.982}_{\pm 0.003}$ | 9.17 |
| PCFI | $0.949_{\pm 0.004}$ | $0.955_{\pm 0.006}$ | 0.57 |
| Mean | $0.979_{\pm 0.006}$ | $0.979_{\pm 0.004}$ | 0.08 |
| kNN | $0.969_{\pm 0.011}$ | $0.977_{\pm 0.005}$ | 0.83 |
| GAIN | $0.978_{\pm 0.011}$ | $0.982_{\pm 0.007}$ | 0.39 |
| MIWAE | OOM | - | - |
| GRAPE | OOM | - | - |
| IGRM | OOM | - | - |
| scFP | $0.952_{\pm 0.004}$ | $0.976_{\pm 0.003}$ | 2.52 |

Table 13: Medical-Duke Breast.

| | Duke Breast (IMR: 11.94%) | | |
|---|---|---|---|
| | OG | + GRASS init. | Impr. (%) |
| LP | $0.672_{\pm 0.021}$ | $0.678_{\pm 0.026}$ | 0.98 |
| GCNMF | $0.664_{\pm 0.035}$ | $0.688_{\pm 0.032}$ | 3.61 |
| PaGNN | $0.685_{\pm 0.033}$ | $0.690_{\pm 0.029}$ | 0.69 |
| Zero | $0.673_{\pm 0.022}$ | $0.694_{\pm 0.021}$ | 3.13 |
| NM | $0.678_{\pm 0.033}$ | $0.691_{\pm 0.025}$ | 1.96 |
| FP | $0.661_{\pm 0.031}$ | $\underline{0.688}_{\pm 0.028}$ | 4.21 |
| PCFI | $0.693_{\pm 0.029}$ | $0.696_{\pm 0.030}$ | 0.40 |
| Mean | $0.687_{\pm 0.018}$ | $0.687_{\pm 0.019}$ | 0.04 |
| kNN | $0.692_{\pm 0.026}$ | $0.697_{\pm 0.014}$ | 0.74 |
| GAIN | $0.699_{\pm 0.018}$ | $\mathbf{0.699}_{\pm 0.017}$ | 0.09 |
| MIWAE | $0.692_{\pm 0.013}$ | $0.693_{\pm 0.012}$ | 0.13 |
| GRAPE | OOM | - | - |
| IGRM | OOM | - | - |
| scFP | $0.678_{\pm 0.031}$ | $0.690_{\pm 0.030}$ | 1.76 |

## A.10 COMPLEXITY ANALYSIS

As GRASS serves as a preprocessing step that aligns with existing baselines to enhance their performance, it is crucial to consider its computational demand alongside its performance benefits in two perspectives: **Memory** and **Time**.

- **Memory cost**: From a memory perspective, the primary resource utilized by GRASS is the *feature gradient* ($\overline{\nabla}_{\mathbf{X}} \in \mathbb{R}^{N \times F}$), which plays a supplemental role in constructing a column-wise graph. This feature gradient shares the same shape as the original feature

Table 14: Medical-ADNI.

| | ADNI (IMR: 30.02%) | | |
|---|---|---|---|
| | OG | + GRASS init. | Impr. (%) |
| LP | $0.928_{\pm 0.005}$ | $0.943_{\pm 0.005}$ | 1.56 |
| GCNMF | $0.897_{\pm 0.045}$ | $\underline{0.944}_{\pm 0.004}$ | 5.25 |
| PaGNN | $0.953_{\pm 0.003}$ | $0.955_{\pm 0.003}$ | 0.27 |
| Zero | $0.956_{\pm 0.003}$ | $\mathbf{0.957}_{\pm 0.003}$ | 0.17 |
| NM | $0.955_{\pm 0.003}$ | $0.956_{\pm 0.003}$ | 0.19 |
| FP | $0.955_{\pm 0.003}$ | $0.957_{\pm 0.003}$ | 0.18 |
| PCFI | $0.951_{\pm 0.004}$ | $0.955_{\pm 0.003}$ | 0.46 |
| Mean | $0.939_{\pm 0.002}$ | $0.943_{\pm 0.003}$ | 0.46 |
| kNN | $0.943_{\pm 0.003}$ | $0.943_{\pm 0.004}$ | 0.01 |
| GAIN | $0.937_{\pm 0.003}$ | $0.944_{\pm 0.003}$ | 0.67 |
| MIWAE | OOM | - | - |
| GRAPE | OOM | - | - |
| IGRM | OOM | - | - |
| scFP | $0.953_{\pm 0.003}$ | $0.954_{\pm 0.002}$ | 0.10 |

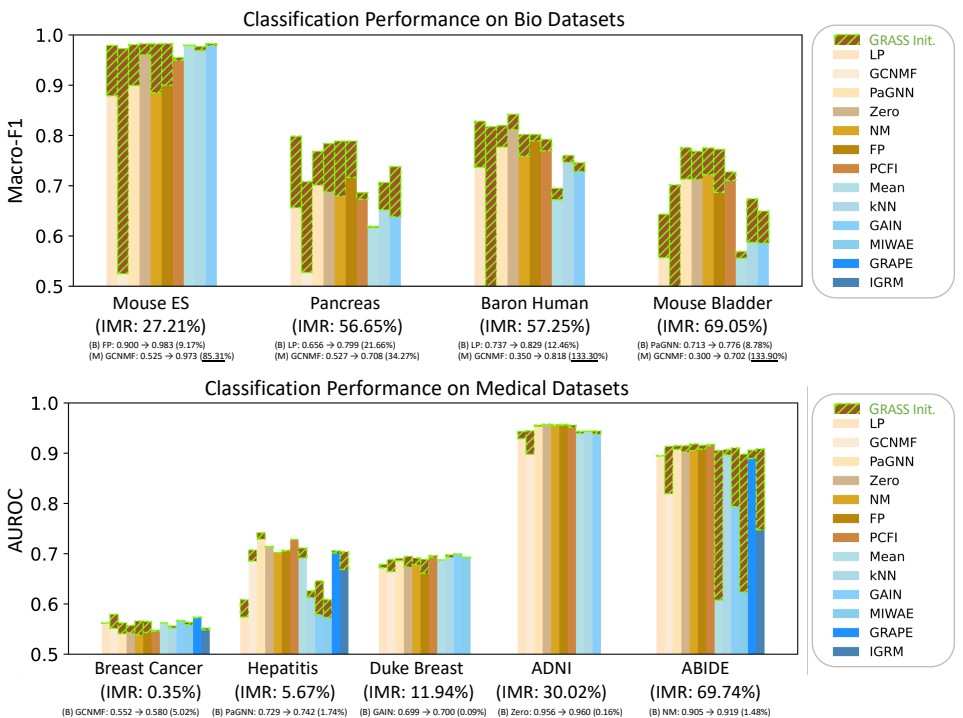

Figure 8: Classification performance on biomedical dataset with initially missing rate (IMR). (B) denotes the "Best" performing baseline while (M) denotes the "Most Improved" baseline with their relative improvement. A percentage is underlined if it surpasses 80%.

matrix, with dimensions corresponding to the total number of nodes ($N$) and features ($F$). However, it is important to note that in the context of graph-based imputation models, which inherently employ a row-wise (sample-wise) adjacency matrix ($A \in \mathbb{R}^{N \times N}$), the complexity associated with the adjacency matrix often surpasses that of the feature matrix,

i.e., $\mathcal{O}(NF) + \mathcal{O}(N^2) = \mathcal{O}(N^2)$. This is particularly true in the bio-medical domain where datasets are typically tabular and the number of samples significantly exceeds the number of features ($N >> F$). Therefore, the additional memory requirement for storing the feature gradient is not prohibitively large. Furthermore, the complexity of the generated column-wise graph ($\mathbf{A}^{feat} \in \mathbb{R}^{F \times F}$) is also lower compared to the row-wise adjacency matrix, allowing GRASS to align with existing graph-based models without incurring excessive memory costs. Once the warmed-up matrix ($\hat{\mathbf{X}}$) and adjacency matrix ($\hat{\mathbf{A}}$) are computed, the memory allocated for the feature gradient and column-wise graph can be released, leaving only the cost of training the original baseline model for the downstream task.

- **Time cost**: From a time complexity perspective, the process is almost identical to training a conventional 2-layer MLP, which is efficient for tabular data and involves training two weight matrices: one that transforms the raw feature space to a hidden space, and another that maps the hidden space to the output space for final predictions. Despite the apparent complexity of calculating the feature gradient as outlined in Proposition 3.2, the actual computation, as demonstrated in Listing 1, is straightforward in terms of implementation. By enabling the 'requires_grad' switch, the gradient information is automatically saved, making the time complexity for computing the feature gradient equivalent to training a 2-layer MLP. Additionally, the column-wise Feature Propagation can be efficiently executed via sparse multiplication of the adjacency matrix and the feature matrix, as detailed in (Rossi et al., 2021). Thus, the overall time required to obtain the warmed-up matrix and adjacency matrix is not substantial.

## A.11 FURTHER EXTENSION AND GENERALIZABILITY OF GRASS

To explore the scalability of GRASS to larger datasets, we conducted evaluations using the single-cell RNA-seq Macosko dataset, which comprises 44,808 cells, 22,452 genes, and 14 distinct cell types, with an initial missing ratio of 81.41%. Among these genes, we preprocessed 2,000 highly variable genes, a common technique in scRNA-seq (Yun et al., 2023). We noted that GRASS integrates smoothly with existing methods, except in cases where initial baselines, such as GCNMF and GRAPE, encounter Out-Of-Memory (OOM) issues due to the weights associated with the Gaussian Mixture Model and the construction of a heterogeneous node-feature graph, respectively. In Table 15, it is observed that graph-based methods can enhance their performance when combined with GRASS. In large graphs, since the feature dimension typically does not surpass the number of samples (which is usually the case), GRASS aligns well with current graph-based imputation methods.

Additionally, while GRASS is primarily designed for the bio-medical domain, we also assessed its applicability to other domains. For this purpose, we utilized the Wine dataset (Asuncion & Newman, 2007), which consists of 178 samples with 14 numerical features and 3 classes. As the Wine dataset initially lacks missing values, we introduced a 30% uniform missing scenario by manually dropping features. Table 16 demonstrates that using GRASS as an initializer, enabling existing models to start with a warmed-up feature matrix and adjacency matrix, effectively benefits other domains as well. This highlights the potential of GRASS for broader generalizability beyond the bio-medical domain.

Table 15: Scalability-Macosko dataset.

| | Macosko (IMR: 81.41%) | | |
|---|---|---|---|
| | OG | + GRASS init. | Impr. (%) |
| LP | $0.853_{\pm 0.025}$ | $\underline{0.870}_{\pm 0.025}$ | 7.19 |
| GCNMF | OOM | - | - |
| PaGNN | $0.938_{\pm 0.008}$ | $0.939_{\pm 0.001}$ | 0.17 |
| Zero | $0.920_{\pm 0.031}$ | $0.929_{\pm 0.006}$ | 0.92 |
| NM | $0.923_{\pm 0.017}$ | $0.930_{\pm 0.071}$ | 0.71 |
| FP | $0.937_{\pm 0.006}$ | $0.941_{\pm 0.045}$ | 0.43 |
| PCFI | $0.932_{\pm 0.017}$ | $0.939_{\pm 0.005}$ | 0.75 |
| Mean | $0.819_{\pm 0.042}$ | $0.835_{\pm 0.048}$ | 1.87 |
| kNN | $0.904_{\pm 0.021}$ | $0.910_{\pm 0.012}$ | 0.62 |
| GAIN | $0.891_{\pm 0.039}$ | $0.898_{\pm 0.012}$ | 0.86 |
| MIWAE | OOM | - | - |
| GRAPE | OOM | - | - |
| IGRM | OOM | - | - |
| scFP | $0.934_{\pm 0.011}$ | $\mathbf{0.941}_{\pm 0.020}$ | 0.72 |

Table 16: Generalizability-Wine dataset.

| | Wine (IMR: 0.00%) | | |
|---|---|---|---|
| | OG | + GRASS init. | Impr. (%) |
| LP | $0.647_{\pm 0.017}$ | $0.647_{\pm 0.017}$ | 0.00 |
| GCNMF | $0.656_{\pm 0.027}$ | $0.657_{\pm 0.023}$ | 0.11 |
| PaGNN | $0.650_{\pm 0.030}$ | $0.661_{\pm 0.023}$ | 1.65 |
| Zero | $0.637_{\pm 0.043}$ | $0.648_{\pm 0.033}$ | 1.68 |
| NM | $0.629_{\pm 0.034}$ | $0.660_{\pm 0.030}$ | 4.93 |
| FP | $0.642_{\pm 0.032}$ | $0.647_{\pm 0.042}$ | 0.79 |
| PCFI | $0.650_{\pm 0.042}$ | $\mathbf{0.670}_{\pm 0.031}$ | 3.06 |
| Mean | $0.585_{\pm 0.028}$ | $0.600_{\pm 0.021}$ | 2.46 |
| kNN | $0.629_{\pm 0.012}$ | $0.640_{\pm 0.012}$ | 1.75 |
| GAIN | $0.618_{\pm 0.012}$ | $0.640_{\pm 0.018}$ | 3.61 |
| MIWAE | $0.514_{\pm 0.027}$ | $\underline{0.591}_{\pm 0.024}$ | 14.93 |
| GRAPE | $0.567_{\pm 0.064}$ | $0.587_{\pm 0.045}$ | 3.52 |
| IGRM | $0.573_{\pm 0.022}$ | $0.579_{\pm 0.044}$ | 0.96 |
| scFP | $0.620_{\pm 0.022}$ | $0.620_{\pm 0.026}$ | 0.10 |

