# OpenReview forum: "Toward Generalizability of Graph-based Imputation on Biomedical Tabular-based Missing Data"
_ICLR.cc/2025/Conference — ICLR 2025 Conference Withdrawn Submission_

### Official Review · Reviewer_RYkj · 2024-11-02

**Soundness:** 2
**Presentation:** 1
**Contribution:** 2
**Rating:** 5
**Confidence:** 2

**Summary:**

The paper proposes a method for missing data imputation by using additional information of gradient information of a MLP with graph-based imputation.

**Strengths:**

It is a combination of several sensible ideas. It might make sense that some of these combinations might give a good performance.

**Weaknesses:**

- The method's description is not clear from the first point of training a MLP.
- The contribution of the paper, and the contrast with previous work, is over-exaggerating, making the contribution difficult to judge.
- The key idea of using gradient information as additional information on the data is not explored. Not sure why is it suitable/good for the problems of biomedical data, missing data.
I think that overall, the paper exaggerate the "underexplored" problems, claiming graph-based methods having "strong inductive bias", while providing no clear evidence of the problems nor showing no signs of the solutions. The paper focuses mainly on these points and lack the evidences/reasonings of why the proposed method should work.

**Questions:**

- How to train the MLP with the missing data?
- What does "we choose not to rely on the initial partially observed feature matrix" mean? MLP is trained on a different dataset?
- What is the "generalizability a graph-based method to tabular data" when there are many methods for generating graphs available, and the paper proposes another one?
- The imputation methods are for regression only? I don't see what is the "underexplored" classification setting is a problem.
- Graph-based methods for biomedical data is underexplored? Are those methods not applicable to this domain?

---

### Official Review · Reviewer_PgB4 · 2024-11-02

**Soundness:** 2
**Presentation:** 1
**Contribution:** 3
**Rating:** 5
**Confidence:** 4

**Summary:**

The authors address a crucial issue - the failure of current message-passing methods in biomedical imputation. While powerful message-passing methods have been developed in graphical data with strong homophily, even with high fractions of missing data (way above 90%), these methods fail in biomedical data, mainly since there is no inherent homophilic graph.
The authors propose to address that by creating a graph based on the gradient of an MLP for the classification task in hand, and show on a few datasets that they improve on the prediction accuracy.

**Strengths:**

The task studied is of crucial importance
The idea of using the gradient (albeit not very new) is important since it produces associations that are task-specific
The literature review is clear
The solution proposed beats the state of the art on the studied datasets.

**Weaknesses:**

The presentation is very poor. It is impossible to understand the methods or the quality of the experiments using the main text only.
The experiments are very limited, with very few experiments.
There is no real discussion of hyperparameters, structure of the network..... except for two parameters K_col and K_row, but even those are very poorly tested

**Questions:**

How were hyperparameters tuned
Can you please clarify in the text how was the matrix A created from the similarity of the gradients

---

### Official Review · Reviewer_96Tm · 2024-11-02

**Soundness:** 2
**Presentation:** 1
**Contribution:** 2
**Rating:** 5
**Confidence:** 4

**Summary:**

The authors propose to tackle the problem of missing data imputation in tabular data by augmenting the features with gradient information from a MLP trained on a predictive task and propagating the non missing information first column-wise and then row-wise.

**Strengths:**

The notion of augmenting feature descriptors using information from the dynamics of learning algorithms is of interest.

**Weaknesses:**

1. since the main focus is the capacity of dealing with missing values it is necessary to show how the approach behaves as the type and level of missingness changes
2. the authors claim that: 'feature gradient plays a crucial supplement role in learning more distinct class representations'. This is an interesting claim that needs to be defended either theoretically or empirically.

**Questions:**

1.a. please provide empirical experimentation on how the amount of missingness at random impacts the performance of the proposed approach
1.b. if possible provide a similar analysis when the missingness is not at random
2. why using gradients?  why not using the embedding representation of the MLP (e.g. last layer, last -1, etc)? please justify the choice. It would improve the paper a direct comparison with an augmentation approach based on the last layers of the MLP.
- please consider reporting results using the critical differences diagram (https://scikit-posthocs.readthedocs.io/en/latest/generated/scikit_posthocs.critical_difference_diagram.html) to show that, averaging across all datasets, each approach is significantly improved by using the GRASS init.
- please consider using \citep in the text rather than \citet

---

### Official Review · Reviewer_9p9K · 2024-11-03

**Soundness:** 2
**Presentation:** 2
**Contribution:** 1
**Rating:** 3
**Confidence:** 4

**Summary:**

This paper proposes an approach for classification on tabular datasets with missing data. The proposed method, GRASS, is designed to generate a graph structure that is used in downstream learning with graph-based imputation methods. GRASS first obtains feature gradients by training an MLP layer and performs column-wise feature propagation. After uncertainty-aware categorical clamping, GRASS builds the graph structure using kNN connections.

**Strengths:**

The motivation of this work is clearly presented, and the paper is well-organized. The authors conducted experiments on various datasets, and the proposed GRASS significantly improves performance on some datasets.

**Weaknesses:**

I believe the comparative experiments are significantly misdesigned. It is well known in tabular data learning that tree-based methods, including CatBoost and XGBoost, are generally more effective than deep learning-based approaches [[1](https://proceedings.neurips.cc/paper_files/paper/2022/hash/0378c7692da36807bdec87ab043cdadc-Abstract-Datasets_and_Benchmarks.html), [2](https://doi.org/10.1016/j.inffus.2021.11.011)]. Comparisons with methods known to perform well in tabular data classification are entirely omitted. A comparison table with those methods should be included to substantiate the claim that GRASS performs well in the biomedical tabular domain. At the very least, I believe neural networks specialized for classification on tabular data [[3](https://proceedings.neurips.cc/paper_files/paper/2021/hash/9d86d83f925f2149e9edb0ac3b49229c-Abstract.html)] should have been included in the comparison. Isn’t this paper aiming to argue that graph-based imputation using GRASS, which incorporates GNN in the process, is effective for classification on tabualar data containing missing features? If so, graph-based imputation using GRASS should be compared to the current state-of-the-art methods in tabular data learning. The performance when commonly used data imputation and tree-based methods are applied, as well as the performance when using the graph-based imputation using GRASS, should be compared and discussed in this study.

**Questions:**

* Could you conduct experiments to measure the performance when commonly used data imputation and tree-based methods are applied?

* Could you conduct experiments to measure the performance of tree-based methods using graph-based imputation methods with GRASS?

---

### Note · Authors · 2024-12-02

I have read and agree with the venue's withdrawal policy on behalf of myself and my co-authors.